# Peripheral blood levels of brain-derived neurotrophic factor in patients with post-traumatic stress disorder (PTSD): A systematic review and meta-analysis

Helia Mojtabavi[1,2], Amene Saghazadeh[3,4], Leigh van den Heuvel[5], Joana Bucker[6], Nima Rezaei[2,3,4,7] *

1 School of Medicine, Tehran University of Medical Sciences, Tehran, Iran, 2 Systematic Review and Meta-Analysis Expert Group (SRMEG), Universal Scientific Education and Research Network (USERN), Tehran, Iran, 3 Research Center for Immunodeficiencies, Children's Medical Center, Tehran University of Medical Sciences, Tehran, Iran, 4 MetaCognition Interest Group (MCIG), Universal Scientific Education and Research Network (USERN), Tehran, Iran, 5 Department of Psychiatry, Faculty of Medicine and Health Sciences, University of Stellenbosch, Cape Town, South Africa, 6 Bipolar Disorder Program and Laboratory of Molecular Psychiatry, National Institute for Translational Medicine, INCT-TM Hospital de Clínicas de Porto Alegre, Universidade Federal do Rio Grande do Sul, Porto Alegre, Brazil, 7 Department of Immunology, School of Medicine, Tehran University of Medical Sciences, Tehran, Iran

* rezaei_nima@tums.ac.ir, rezaei_nima@yahoo.com

**Data Availability Statement:** All relevant data are within the manuscript and its Supporting Information files.

## Abstract

### Background

Brain-derived neurotrophic factor (BDNF) plays a crucial role in the survival, differentiation, growth, and plasticity of the central nervous system (CNS). Post-traumatic stress disorder (PTSD) is a complex syndrome that affects CNS function. Evidence indicates that changes in peripheral levels of BDNF may interfere with stress. However, the results are mixed. This study investigates whether blood levels of BDNF in patients with post-traumatic stress disorder (PTSD) are different.

### Methods

We conducted a systematic search in the major electronic medical databases from inception through September 2019 and identified Observational studies that measured serum levels of BDNF in patients with PTSD compared to controls without PTSD.

### Results

20 studies were eligible to be included in the present meta-analysis. Subjects with PTSD (n = 909) showed lower BDNF levels compared to Non-PTSD controls (n = 1679) (SMD = 0.52; 95% confidence interval: 0.18 to 0.85). Subgroup meta-analyses confirmed higher levels of BDNF in patients with PTSD compared to non-PTSD controls in plasma, not serum, and in studies that used sandwich ELISA, not ELISA, for BDNF measurement. Meta-regressions showed no significant effect of age, gender, NOS, and sample size.

**Funding:** The author(s) received no specific funding for this work.

**Competing interests:** The authors have declared that no competing interests exist.

## Conclusions

PTSD patients had increased serum BDNF levels compared to healthy controls. Our finding of higher BDNF levels in patients with PTSD supports the notion that PTSD is a neuroplastic disorder.

## Introduction

Brain-derived neurotrophic factor (BDNF), as the richest neurotrophin in the brain, was initially described for its role in the central nervous system (CNS) development. It can participate in neural activities, including survival, differentiation, growth, and neuronal plasticity [1]. Because of its rise in response to brain insults, studies suggest an important role for BDNF in neurogenesis [2]. It maintains synaptic plasticity, which is essential in extinction learning and consolidation of fear memories [3, 4].

Post-traumatic stress disorder (PTSD) is *"a psychiatric disorder that can occur in people who have experienced or witnessed a traumatic event such as a natural disaster, a serious accident, a terrorist act, war/combat, rape or other violent personal assault"* [5]. PTSD corresponds to a lifetime prevalence of 7.8% in the USA [6]. Its clinical picture varies from re-experiencing, including nightmares, intrusive thoughts and flashbacks of the trauma, and avoidance of the remainders of trauma, to hyperarousal such as exaggerated startle response, sleep disturbances, and impaired learning and concentration. Several brain areas are supposed to be involved in the pathophysiology of PTSD, including the hippocampus, amygdala, and cingulate alongside the medial and dorsolateral prefrontal cortex [7].

The neurobiology of PTSD has not been fully elucidated [8]. However, studies are consistent that individuals with PTSD suffer from excessive consolidation of fear memories and extinction learning [4]. These specific features of PTSD are found to be associated with changes in BDNF levels [9]. Changes in BDNF levels are marked in neuropsychiatric disorders, in particular, depression [10]. Although most studies pose a decline in the BDNF values post-stress [7, 11, 12], others fail to report this pattern of association [13].

We performed the present systematic review and meta-analysis to investigate whether peripheral levels of BDNF are different in people with PTSD.

## Materials and method

### Literature search and selection criteria

Relevant studies were identified by searching keywords (post-traumatic stress disorder OR PTSD) and (Brain-Derived Neurotrophic Factor OR BDNF) on PubMed and Scopus in September 2019. To avoid missing potential articles, the reference list of all relevant articles previously identified through electronic searching was checked.

We included original articles meeting the following criteria; (1) an observational study to measure serum or plasma levels of BDNF in PTSD population and control subjects without PTSD (2) providing sufficient data including total subjects' population and mean and standard deviation (SD) of the BDNF levels for each study group.

### Data extraction

We extracted the following data from each included publication; first-named author, year of publication, location of study, the assay that was used for BDNF measurement, type of specimen taken from subjects, number of subjects in the PTSD population and control group,

demographic characteristics (e.g., age and gender) of both groups, mean ± SD of the BDNF levels, and the measurement scale (e.g., pg/mL, ng/mL, or ng/mg) of BDNF levels.

Data was either extracted from the manuscript, converted from the provided tables or corresponding authors were contacted and invited to share their results in case the prior two approaches were unsuccessful. An excel spreadsheet containing details of extracted data is available on request.

### Quality assessment

Newcastle–Ottawa scale (NOS) designed for observational studies was applied to assess the quality of included articles [14]. The NOS rates observational studies by three main aspects; sample selection, comparability of cases and controls, and exposure. The possible score range in NOS is from 0 to 9; studies with scores of 7–9 stars represent the highest quality with the lowest risk of bias, while studies scoring below four stars have the highest risk of bias and the lowest quality. The remaining studies that fall within 4 to 6 stars have a moderate risk of both bias and quality.

### Quantitative analysis

All of the statistical analyses were done by Review Manager (RevMan) Version 5.3 [15]. We used fixed effects and random effects model to analyze the continuous data, mean and SD of BDNF levels, on the outcome of our two study groups. Based on Cochrane guidelines, we determined heterogeneity by means of Q statistic tests and the $I^2$ index. We intended to use the fixed effects model in this meta-analysis. Since an $I^2$ value of more than 40% indicates inconsistency across studies, we planned to switch to the random effects model in the case of $I^2$ fluctuated more than 40%. To measure the effect, we used a standardized mean difference (SMD) in studies using different measurement scales or assays. Otherwise, the mean difference (MD) was extracted for the effect measurement. The risk of publication bias was assessed using the degree of funnel plot asymmetry. We considered a *p*-value of less than 0.05 statistically significant.

## Results

### Study selection

Searching the database revealed a total number of 350 records. Preliminary screening excluded 81 duplicated records and 209 studies based on title and abstract. The full text of sixty articles was assessed carefully based on inclusion and exclusion criteria. A number of 27 records did not measure the BDNF level. We excluded 13 additional studies due to the following reasons: four articles for being non-diagnostic [16–19], two for not being original studies [20, 21], and the other two due to insufficient data [12, 22]. Five more studies were also excluded due to measuring BDNF in CSF [23], investigating on small sample size [24], using animal samples [25], using replicated data [12], and having no control group [26]. Finally, a total number of 20 articles entered our meta-analysis. Fig 1 clearly illustrates the study selection.

### Study characteristics

A total number of 911 subjects diagnosed with PTSD were included in 20 independent studies. The PTSD population consisted of 35.9% male, with a mean age of 37.85 years (9.44–87.5 years). In the control group, 45.43% of 1689 healthy subjects were male participants with a mean age of 34 years (8.96–44.5 years).

The total study population had different clinical backgrounds such as childhood sexual abuse [8, 27, 28], antepartum PTSD [29], earthquake survivors [30], road traffic accident

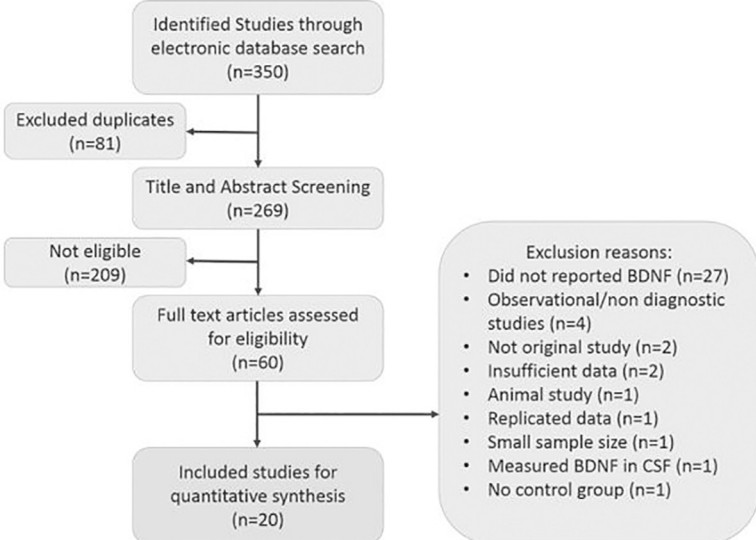

**Fig 1. PRISMA flow diagram.**

induced PTSD [31–33], war zone related PTSD [6, 34], and also in the presence of other psychiatric disorders like depression [35], bipolar disorder (BD) [36], and alcohol abuse [37] or severe medical conditions like Hepatocellular carcinoma (HCC) [38]. Although our included patients had a diverse prior clinical background, all of them fulfill the definite PTSD criteria. Also, their trauma was scored based on severity in most studies using validated trauma scores such as the Davidson Trauma Scale (DTS) [39]. All of the included studies confirmed the diagnosis of PTSD, mostly using Structured Clinical Interview for DSM-IV axis I disorders (SCID-I). The clinician administered the PTSD scale (CAPS) in their interview, with a cutoff score of 40, which was the most common questionnaire administered in our included studies [6–8, 11, 28, 30, 31, 40, 41]. Most studies did not declare whether the healthy controls had any trauma exposures or not except for Angelucci et al. and Stratta et al. who recruited their controls from trauma-exposed groups who did not develop PTSD [11, 42].

Several ELISA kits were used in these 20 studies; most of the included studies used the Quantikine ELISA kit (7 studies). Chemicon, Promega, Mallipore ChemiKine and Milliplex, Picokine, Raybiotech, Human BDNF ELISA, and Nanjing Jiancheng ELISA kits for human BDNF were the other used kits in the studies. Mallipore Milliplex had the broadest range of detection (12–50000 pg/ml), whereas Promega had a close range of BDNF detection (7.8–500 pg/ml). Also, Mallipore Milliplex was the most sensitive kit among all used with detecting 2.5 pg/ml of BDNF. Lastly, Mallipore Milliplex and Quantikine from R&D were the top two kits regarding higher intra-assay CV and inter-assay CV among all [43].

In summary, except for Dotta-Panichi et al. [13], Hauck et al. [39], Martinotti et al. [40], and Matsuoka et al. [31], who used sandwich enzyme-linked immunosorbent assay (ELISA), remaining included studies used conventional ELISA to measure BDNF. The characteristics of the included studies have been summarized in Table 1. The quality of the included studies [14] was ranged from 6 to 8, with a mean NOS value of 7.1. Detailed scores can be found in Table 2.

## A meta-analysis of blood BDNF levels

Twenty studies measured the BDNF levels in the blood specimens (either serum or plasma) of the PTSD population (n = 909) and control subjects (n = 1679). There was significant

**Table 1. Demographic characteristics of the included studies.**

| First author, Year | Location | Sample | PTSD | | | Control | | |
|---|---|---|---|---|---|---|---|---|
| | | | No. | Male % | Mean age | No. | Male % | Mean age |
| Aksu, 2018 [8] | Turkey | Serum | 28 | 0 | 14.7 | 31 | 0 | 14.7 |
| Angelucci, 2014 [11] | Italy | Serum | 23 | 52.17 | 40.8 | 19 | 57.89 | 37.3 |
| Blessing, 2017 [6] | USA | Serum | 79 | 100 | 33.0 | 78 | 100 | 32.5 |
| Bucker, 2015 [28] | Brazil | Plasma | 36 | 61.11 | 9.44 | 26 | 57.69 | 8.96 |
| Dell'Osso, 2009 [7] | Italy | Plasma | 18 | 33.33 | 42.1 | 18 | 38.88 | 38.8 |
| Dotta-Panichi, 2015 [13] | Brazil | Serum | 18 | 0 | 34.39 | 36 | 0 | 35.22 |
| Grassi-Oliveira, 2008 [35] | Brazil | Plasma | 17 | 0 | 39.35 | 15 | 0 | 36.47 |
| Guo, 2019 [38] | China | Serum | 102 | 41.17 | 44.00 | 298 | 64.7 | 43.98 |
| Hauck, 2010 [39] | Brazil | Serum | 13 | 20.58 | 35.2 | 34 | 20.58 | 36.2 |
| Kauer-Sant'Anna, 2007 [36] | Brazil | Serum | 78 | 25.0 | 42.13 | 85 | 31.1 | 43.01 |
| Martinotti, 2015 [40] | Italy | Serum | 20 | 52.17 | 40.9 | 18 | 57.89 | 37.3 |
| Matsuoka, 2013 [31] | Japan | Serum | 8 | 37.5 | 41.3 | 85 | 74.11 | 36.3 |
| Neupane, 2017 [37] | Nepal | Serum | 32 | 33.1 | 87.5 | 154 | 89.7 | 35.9 |
| Simsek, 2015 [27] | Turkey | Serum | 27 | 25.92 | 14.9 | 28 | 35.71 | 13.9 |
| Stratta, 2016 [42] | Italy | Plasma | 13 | 30.0 | 44.9 | 14 | 14.28 | 44.5 |
| Su, 2015 [32] | China | Plasma | 11 | 45.45 | 40.4 | 19 | 63.15 | 40.2 |
| Tural, 2018 [30] | Turkey | Serum | 15 | 0 | 42.53 | 13 | 0 | 43.31 |
| van den Heuvel, 2016 [41] | South Africa | Plasma | 10 | 90 | 33.9 | 80 | 57.5 | 33.46 |
| Yang, 2016 [29] | Peru | Serum | 332 | NA* | NA | 601 | NA | NA |
| Zhang, 2014 [34] | USA | Plasma | 31 | NA | NA | 37 | 100 | NA |

NA = not available

heterogeneity across studies, and therefore, the random-effects model of analysis was used for effect size estimation. BDNF levels were significantly higher in the PTSD population compared to controls with the SMD of 0.52 (95% confidence interval: 0.18 to 0.85, p = 0.003) (Fig 2). The effect was observed irrespective of the control type, e.g., healthy controls or controls without PTSD. Subgroup meta-analyses confirmed higher levels of BDNF in patients with PTSD compared to non-PTSD controls in plasma, not serum, and in studies that used sandwich ELISA, not ELISA, for BDNF measurement (Figs 3 and 4). Meta-regressions showed no significant effect of age, gender, NOS, and sample size. By specifying I2 = 10% and tau2 = 0.25, sensitivity meta-analyses were also performed to check the impact of heterogeneity on effect size. In both cases, the effect size remained significant (tau2 = 0.25: Hedges's g, 0.53, p = 0.000; I2 = 10%: Hedges's g, 0.57, p = 0.000). No evidence of publication bias was found (Fig 5; Egger's p = 0.629; Begg's p = 0.284).

## Discussion

The present study includes data on 909 participants diagnosed with PTSD and 1679 non-PTSD controls, extracted from twenty case-control studies. A meta-analysis was conducted on all the included studies. As noted in Fig 2, the meta-analysis showed significantly higher blood BDNF levels in the PTSD population than control subjects.

BDNF is a protein belonging to the neurotrophins class. Alongside its receptor, tropomyosin receptor kinase B (TrkB), BDNF serves as a survival factor for selected populations of neurons. Loss of proper production or utilization of this protein can result in various CNS disorders [8, 44]. The molecule primarily translates as proBDNF, which is converted to the

Table 2. Quality assessment of studies included in the quantitative synthesis.

| First author, Year | Selection | | | | Comparability | | Exposure | | | Score |
|---|---|---|---|---|---|---|---|---|---|---|
| Aksu, 2018 [8] | * | * | * | * | * | * | * | * | * | 8 |
| Angelucci, 2014 [11] | * | * | - | - | * | * | * | * | * | 6 |
| Blessing, 2017 [6] | * | * | - | * | * | * | * | * | * | 7 |
| Bucker, 2015 [28] | * | * | * | * | * | * | * | * | * | 8 |
| Dell'Osso, 2009 [7] | * | * | * | * | * | * | * | * | * | 8 |
| Dotta-Panichi, 2015 [13] | * | * | * | * | * | * | * | * | * | 8 |
| Grassi-Oliveira, 2008 [35] | * | * | * | * | * | * | * | * | * | 8 |
| Guo, 2019 [38] | * | * | - | - | * | * | * | * | * | 6 |
| Hauck, 2010 [39] | * | * | * | * | * | * | * | * | * | 8 |
| Kauer-Sant'Anna, 2007 [36] | * | * | - | - | * | * | * | * | * | 6 |
| Martinotti, 2015 [40] | * | * | - | * | * | * | * | * | * | 7 |
| Matsuoka, 2013 [31] | * | * | - | * | * | * | * | * | * | 7 |
| Neupane, 2017 [37] | * | * | - | - | * | * | * | * | * | 6 |
| Simsek, 2015 [27] | * | * | - | - | * | * | * | * | * | 6 |
| Stratta, 2016 [42] | * | * | - | * | * | * | * | * | * | 7 |
| Su, 2015 [32] | * | * | - | * | * | * | * | * | * | 7 |
| Tural, 2018 [30] | * | * | - | * | * | * | * | * | * | 7 |
| van den Heuvel, 2016 [41] | * | * | - | * | * | * | * | * | * | 7 |
| Yang, 2016 [29] | * | * | * | * | * | * | * | * | * | 8 |
| Zhang, 2014 [34] | * | * | - | * | * | * | * | * | * | 7 |

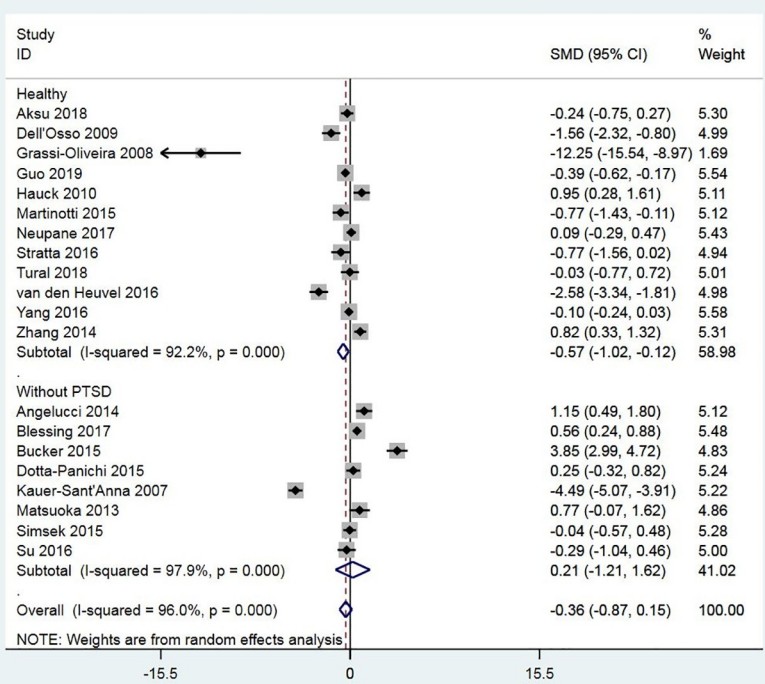

Fig 2. A meta-analysis of BDNF levels in PTSD patients compared to non-PTSD controls by the type of control (healthy controls and patients without PTSD).

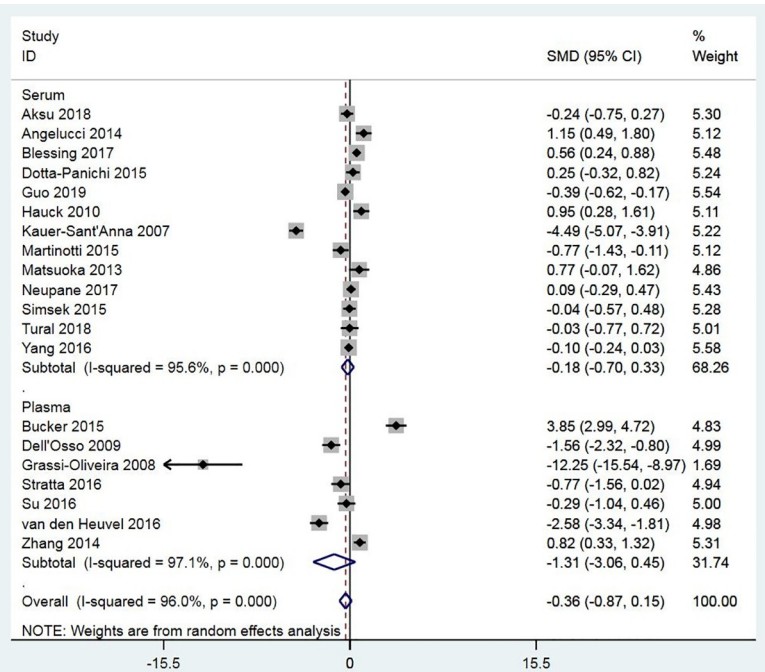

**Fig 3. A meta-analysis of BDNF levels in PTSD patients compared to non-PTSD controls by the technique used for BDNF measurement (ELISA and sandwich ELISA).**

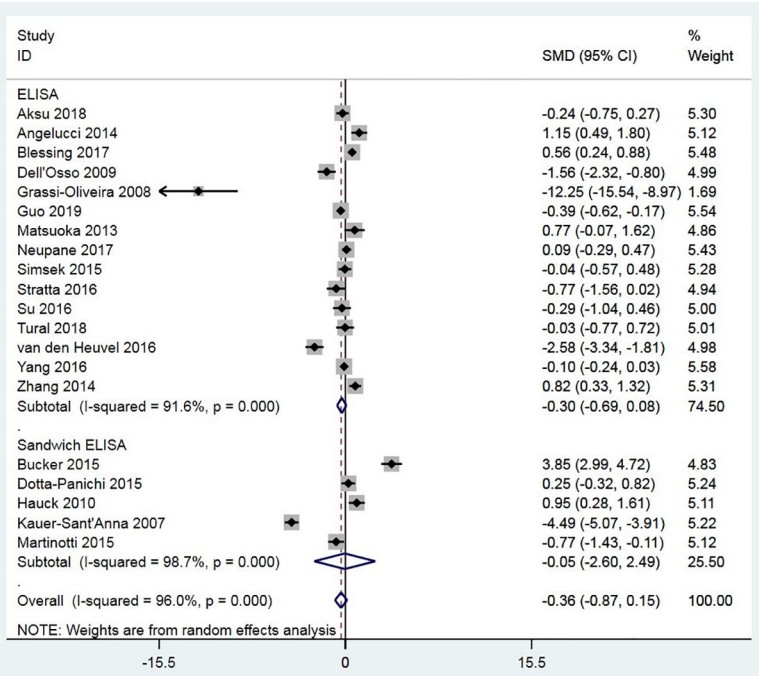

**Fig 4. A meta-analysis of BDNF levels in PTSD patients compared to non-PTSD controls by the specimen used for BDNF measurement (serum and plasma).**

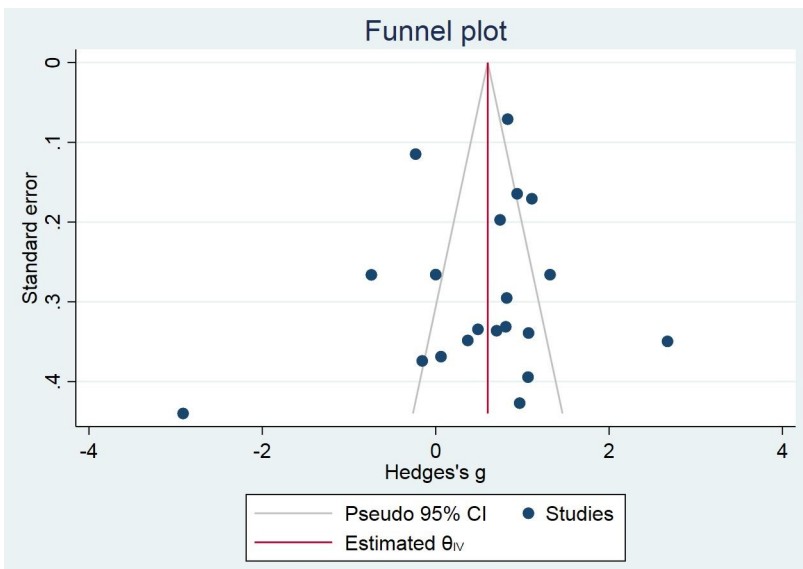

**Fig 5. Funnel plot for meta-analysis related to BDNF levels in PTSD patients compared to non-PTSD controls.**

mature BDNF through tissue plasminogen activator (tPA). Contrary to its mature form, which promotes cell survival and neuronal plasticity, proBDNF functions as a proapoptotic molecule [45]. Neuroinflammation is known to affect several BDNF-related signaling pathways causing different brain pathologies. Aging is among the most affected brain changes by the downregulation of neurotrophic factors. BDNF is also found to be associated with the pathogenesis of several neuropsychiatric and neurodegenerative disorders, including major depressive disorder (MDD), bipolar disorder (BD), schizophrenia, Parkinson disease, Alzheimer disease, and epilepsy [46–49]. It is expected that presenting a particular mental illness relies on the combination of environmental, genetic, and temperamental factors [50].

In regards to the underlying depression as a confounder variable, attitudes were diverse in our selected studies. Some papers excluded subjects with depression to dismiss the potential effect of this disorder on the BDNF level [7, 11, 40]. However, others either did not exclude the depressed samples [8, 29] or specifically investigated the correlation of BDNF levels in PTSD patients in an MDD population [35]. In the study by Aksu et al. [8], measured levels of BDNF, proBDNF, and tPA showed no difference between the PTSD group with depressive symptoms (21%) and the remaining PTSD population. Although 46 out of 83 cases (55%) in Blessing et al.'s study had a confirmed diagnosis of MDD in addition to the PTSD and twenty of them consume anti-depressant drugs, no correlation analysis was carried out to address the effect of this factors on BDNF measures [6]. Despite detecting symptoms of depression, mania, and anxiety with or without drug consumption, Bucker et al. and Dotta-Panichi et al. did not include these variables in their analysis [13, 28].

On the other hand, recurrent MDD patients were investigated to address the association of BDNF level and trauma exposure in the presence of MDD as a major comorbidity by Grassi-Oliveira et al. They concluded that patients with MDD had lower levels of BDNF compared to the healthy control population and within the MDD group the ones with the presence of childhood physical neglect (CPN), as a predisposing factor for PTSD, observed to have even lower BDNF measures [35]. Although 52.9% of MDD with CPN group and 41.2% of MDD without CPN group consumed selective serotonin reuptake inhibitors (SSRI) as an anti-depressant agent, the study did not control the outcome to address this issue [35]. Anti-depressant

consumption, current or past psychiatric disorder, or depressive symptoms had zero effect on the BDNF level in the sample studied by Hauk et al. [39]. Ultimately, in the study by Yang et al., a conclusion was made that women with comorbid PTSD-depression have 1.52-fold increased odds of having lower levels of serum BDNF compared to the women with neither of these conditions. Nevertheless, the difference between BDNF measures for the two mentioned groups did not reach a level of significance despite the lower BDNF amount in the PTSD-depression group [29]. Despite SSRI or other psychoactive drug consumption in some of our included studies, the effect of medication was not analyzed in any of the studies.

Moreover, in patients with BD, BDNF levels significantly decreased during both manic and depressive episodes as compared with patients in remission (euthymic) and with healthy controls [51]. In the study by Kauer et al. [36], investigating 163 patients with BD, no difference was found between the groups with and without trauma exposure regarding anti-depressants or antipsychotics consumption. Also, BDNF levels were significantly lower in BD patients with a history of trauma, which is similar to previous studies [36].

Low serum BDNF levels have been reported among schizophrenia and in suicidal subjects, while SSRIs, mood stabilizers, and electroconvulsive therapy correlate with higher BDNF levels [40]. BDNF has a remarkable role in learning and motivation [2]. Additionally, the maintenance of dopaminergic and cholinergic neurons relies remarkably on this protein [52–54]. According to a study done by Chen et al., a variant of BDNF can also participate in genetic predispositions to anxiety disorders [55]. In conclusion, nine studies out of 20 included studies detected depressive symptoms with or without anti-depressant consumption. Nonetheless, it is still unclear to what extent PTSD can attribute to the lower levels of measured BDNF in the presence of depression and SSRI intake.

Standard ELISA kits lack proBDNF antibodies and are unable to distinguish between proBDNF and mature BDNF in the bloodstream. New generations of ELISA kits provide antibodies for both pro and mature BDNF. In most included studies in the present meta-analysis, only BDNF was measured, except for the study by Aksu et al., who measured proBDNF in addition to BDNF and tPA. Aksu concluded that PTSD population had lower levels of BDNF and its precursor, proBDNF, compared to their healthy volunteers [8]. The possible limitation of the present study is the absence of proBDNF measures in the remaining studies rather than Aksu [8]. Based on prior investigations on the role of BDNF among people with MDD, although BDNF may be significantly lower in the disease group compared to the normal controls, the difference of proBDNF levels between MDD patients and the healthy population was not significant [65].

The BDNF stress-sensitivity hypothesis describes that BDNF plays a role in mediating environmental factors on vulnerability to stress and related disorders such as PTSD and trauma [56]. Glucocorticoid receptors (GRs) and TrkB are chiefly responsible for mediating such an effect of BDNF in the brain. GRs are widely distributed over the brain regions that underlie the effect of cortisol in stress homeostasis as well. Therefore, the effect of both BDNF and cortisol on stress occurs at these receptors in the brain. By short activation of GR, BDNF can exert a long-lasting effect on memory consolidation. BDNF interacts with its receptor, TrkB, causing its phosphorylation and its downstream signaling pathways. Recent studies propose alterations in BDNF—TrkB signaling as the underlying pathology to some PTSD-related manifestations such as intrusive and incomplete memories, hyperarousal, fear expression, and restricted range of effect [9].

Stress can alter BDNF expression and impair consolidation and reconsolidation mechanisms, and this impairment is common to both acute and chronic stress conditions. BDNF has a remarkable role in learning, memory consolidation, and motivation [2]. Additionally, BDNF is necessary for the maintenance of dopaminergic and cholinergic neurons [52–54]. Saruta et al. noted a significant increase in plasma BDNF concentration in rats following acute immobilization stress [57]. In contrast, chronic exposure to dexamethasone can suppress BDNF-

induced glutamate release, which affects the processes of long-term memory consolidation and developing mental illnesses.

BDNF is involved in the pathogenesis of PTSD and related memory impairments. PTSD is the prototypical form of stress-induced mental disorder [42, 50, 58]. The most fundamental data on the relation of stress exposure and PTSD to BDNF come from animal models of chronic stress [59]. These models suggest that chronic stress causes lower BDNF levels in the hippocampus and prefrontal cortex, leading to the atrophy of the hippocampus and ventrome-dial prefrontal cortex. It can also manifest as deficits in extinction learning [59]. Further, patients with PTSD are more likely to recall negative memories than control subjects [60]. Studies have shown the association of decreased BDNF expression with negative memory bias in patients with PTSD [60].

Hauck et al. concluded from their study that patients with PTSD and ASD caused by recent trauma had higher levels of BDNF compared to the controls in the early years after the trauma, while BDNF downgrades in the PTSD patients with long-term PTSD, four years for instance [39]. The association of BDNF and time past from trauma follows a descending pattern [39]. This raises the question of whether the interval between trauma exposure and sampling for BDNF affects the levels of BDNF or any of its precursors. Another study conducted by Mat-suoka et al. [31] inquired the effect of time on BDNF concentration among people experienc-ing RTA, right after the accident and after 6-months. Results suggest that serum BDNF level was higher after a 6-months follow-up compared to their baseline values. However, this could be considered as acute phase modifications of the brain to maintain its plasticity after a trau-matic experience [31]. Since the time between trauma occurrence and blood sampling for BDNF differed a lot among the studies, we did not include this variable in our analysis, although it could influence the BDNF level significantly.

BDNF genetic variants correlate with abnormal brain structure in PTSD. The gene encod-ing BDNF protein is located at 11p14.1 [61]. BDNF polymorphism rs6265, also known as Val66Met, is the most common single nucleotide polymorphism (SNP) of this gene. This SNP results in a substitution of methionine (Met) for valine (Val) at codon 66 in the pro-domain of the human BDNF protein [62]. The expression of the BDNF gene is altered in fear condition-ing and extinction. Fear conditioning, regarded as associative learning, takes place in the baso-lateral amygdala. Ventromedial prefrontal cortical, which is traditionally believed to inhibit the activity in the amygdala, mediates new inhibitory learnings known as fear extinction [63]. Human studies have frequently shown a correlation between decreased BDNF expression and increased Met allele of the BDNF Val66Met polymorphism in the PTSD population [60]. However, the evidence is not conclusive about the effect of BDNF Val66Met polymorphism on PTSD as a recent meta-analysis of 11 studies estimated a marginal effect of this polymorphism on the risk of PTSD [64]. It is noteworthy to mention that multivariate analyses indicate that the presence of the serotonin transporter (5-HTTLPR) variant further strengthens the correla-tion between BDNF Val66Met polymorphism and the risk of PTSD in adults [65]. Also, Met allele carriers have impaired fear extinction and decreased hippocampal volume and func-tion that are all observed in individuals with PTSD [66].

Also, there is evidence that BDNF genetic variants contribute to the association of child-hood trauma with the risk of developing PTSD in adulthood. The study by Jin et al. [67] reported that people with BDNF Val66Val polymorphism and higher scores for childhood trauma are more likely to develop more severe PTSD symptoms. Interestingly, these people showed a ticker cortex within the left fusiform and transverse temporal gyri and displayed more psychological symptoms, e.g., anxiety, depression, and rumination.

Three papers evaluated the effect of genetic polymorphism on the BDNF plasma levels in the PTSD setting. In the study by van den Heuvel et al. [41] Met allele was associated with a

higher incidence of acute stress disorder (ASD), while after six months follow-up, the prevalence of PTSD was lower in the Met allele cases versus the Val allele group (one and 14 patients respectively). However, none of the observed alterations were statistically significant. Zhang et al. [34] showed that patients with a Met/Met genotype had a higher PTSD symptom severity, while Val carriers had lower PTSD symptom severity. Also, Met carriers had a higher level of BDNF levels in the PTSD group. Lastly, zero association was noted in regards to BDNF G11757 and rs6265 polymorphisms and PTSD severity in the study by Guo et al. [38]. Although the included studies have concluded the significant effect of BDNF among their study population, they have also failed to adequately address the interaction of genetic polymorphisms, the synergic effect of other genes, and the influence of the environment on the previous domains.

Both pharmacologic and non-pharmacologic interventions have been shown to alter BDNF expression associated with PTSD symptom improvement. Ketamine has shown a promising effect on alleviating PTSD symptoms in some animal studies [68]. Acting as a nonselective N-methyl D-aspartate (NMDA) receptor antagonist, ketamine has anti-depressant like effects at subanesthetic doses [69]. Hyperpolarization activated cyclic nucleotide-gated potassium channel (HCN) family which are fundamental in several important neuronal functions such as cellular excitability, dendritic integration, synaptic transmission, neuroplasticity, and rhythmic activity could also be inhibited by ketamine [70]. HCN appears to promote BDNF. It is, however, poorly conceived that ketamine could increase BDNF synthesis through HCN among the PTSD population [68]. Moreover, physical activity might promote hippocampal neurogenesis, size, and function, and help to alleviate anxiety, depression, and PTSD. A possible mechanism of action through which exercise acts is by increasing BDNF levels [66].

One major limitation of our study is that all the included papers evaluated the BDNF levels in the peripheral bloodstream. It is still unknown to what extend the peripheral levels correspond to the CNS levels of BDNF. Many confounding factors can potentially alter levels of BDNF regardless of traumatic experience. MDD, bipolar, SSRI consumptions are common instances that should be considered in future study designs. Almost all the studies, except one, did not measure proBDNF as a potentially involved biomarker in the disease course. BDNF gene polymorphism should also be considered in future study designs. It is noteworthy to point out that ELISA kits mostly can consider mature BDNF and proBDNF as distinct in a limited sense [71]. Using new ELISA kits which can measure mature BDNF with higher specificity is recommended as another line of future research [72].

The present meta-analysis was conducted to investigate whether serum levels of BDNF among the PTSD population are different from those in non-PTSD controls. Analyzing the data from twenty independent studies revealed that people experiencing PTSD had higher levels of BDNF in their blood samples in comparison to controls. It supports the notion that PTSD is a neuroplastic disorder associated with changes in neurotrophins, in particular, BDNF. Further investigations are required to address the impact of trauma and its severity on BDNF levels in people with PTSD.

## Supporting information

**S1 Checklist.**
(DOC)

## Author Contributions

**Conceptualization:** Amene Saghazadeh.

**Data curation:** Helia Mojtabavi, Amene Saghazadeh, Leigh van den Heuvel, Joana Bucker.

**Formal analysis:** Amene Saghazadeh.

**Investigation:** Helia Mojtabavi.

**Methodology:** Amene Saghazadeh.

**Supervision:** Nima Rezaei.

**Validation:** Helia Mojtabavi, Amene Saghazadeh, Leigh van den Heuvel, Joana Bucker, Nima Rezaei.

**Visualization:** Helia Mojtabavi, Amene Saghazadeh, Leigh van den Heuvel, Joana Bucker, Nima Rezaei.

**Writing – original draft:** Helia Mojtabavi.

**Writing – review & editing:** Helia Mojtabavi, Amene Saghazadeh, Leigh van den Heuvel, Joana Bucker, Nima Rezaei.

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
