## [Decision Letter · Decision Letter 0]

21 Jul 2020

PONE-D-20-17402

Peripheral Blood Levels of Brain-Derived Neurotrophic Factor in Patients with Post-traumatic Stress Disorder (PTSD): A Systematic Review and Meta-analysis

PLOS ONE

Dear Dr. Rezaei,

Thank you for submitting your manuscript to PLOS ONE. After careful consideration, we feel that it has merit but does not fully meet PLOS ONE’s publication criteria as it currently stands. Therefore, we invite you to submit a revised version of the manuscript that addresses the points raised during the review process.

We look forward to receiving your revised manuscript.

Kind regards,

Kenji Hashimoto, PhD

Academic Editor

PLOS ONE

Additional Editor Comments:

I would like to suggest that the following articles should be discussed in the limitation of the discussion.

ELISA kits used in the study can recognize both BDNF and its precursor proBDNF because of lack of selectvity of antibodies used in the kits. Therefore, BDNF levels in the human blood are total BDNF and proBDNF. Please discuss the following articles.

Yoshida T, et al. PLOS ONE 2012; 7(8): e42676.

Hashimoto K. Eur Arch Psychiatry Clin Neurosci 2016; 266(3): 285-287.

https://www.sciencedirect.com/science/article/abs/pii/S0278584609001286?via%3Dihub

In your revision ensure you cite all your sources (including your own works), and quote or rephrase any duplicated text outside the methods section.

Further consideration is dependent on these concerns being addressed.

3. Please remove your figures from within your manuscript file, leaving your figures uploaded only as individual TIFF/EPS image files, uploaded separately.  These will be automatically included in the reviewers’ PDF.

Reviewers' comments:

Reviewer's Responses to Questions

**Comments to the Author**

1. Is the manuscript technically sound, and do the data support the conclusions?

Reviewer #1: Partly

Reviewer #2: No

Reviewer #3: Yes

2. Has the statistical analysis been performed appropriately and rigorously? 

Reviewer #1: Yes

Reviewer #2: I Don't Know

Reviewer #3: No

3. Have the authors made all data underlying the findings in their manuscript fully available?

Reviewer #1: Yes

Reviewer #2: No

Reviewer #3: Yes

4. Is the manuscript presented in an intelligible fashion and written in standard English?

Reviewer #1: Yes

Reviewer #2: Yes

Reviewer #3: Yes

5. Review Comments to the Author

Reviewer #1: This is an interesting and ambitious meta analysis of the association between peripheral BDNF and PTSD. There are several, mostly methodological issues that need clarifications in order to properly assess the conclusions.

The authors need to define how the diagnosis of PTSD were determined in the meta analysis. In the original 20 studies, there is a mixture of clinically diagnosed PTSD, and diagnosis based on scores on PTSD rating scales. For examples, what cut-off scores were used in the various papers to classify a person a suffering from PTSD. How was exposure defined? To what degree were prior trauma, e g adverse childhood experiences accounted for? And, what scales were used, apart from clinical diagnostic interviews, to determine a person had met exposure criteria required at the time for the dx of PTSD. There is no info as to the time between the critical exposure event and the diagnosis of PTSD, nor time between diagnosis and the respective study date. Looking at the original studies, there peripheral concentration of BDNF various in both directions (higher/lower) comparing PTSD cases with controls. In most cases, the 95% confidence interval overlaps with 1, that is, the risk ratio does not significantly differ from the null hypothesis. More info is needed as to how the NOS scores were assigned since these scores are critical because they appear to significantly change the weighting of studies and thus overall outcomes. The authors relate trauma exposure to increased neuroplasticity and memory consolidation. There is a need to expand the Discussion section in terms of why one would expect lower levels of BDNF in PTSD patients - apart from referring to SNP and other pre-exposure changes. In terms of the hippocampus, well-controlled studies, e g involving twins where one has been exposed to trauma and develops PTSD, that there are no differences in hippocampal volume.

Overall, a potentially interesting study. However, the reader would benefit from more detailed information as to methodological approaches used in terms of a comparison across studies with very different design and study participants.

Reviewer #2: The authors examined the difference in the blood levels of BDNF between patients with PTSD and healthy subjects using the meta-analysis. They showed that the BDNF levels in patients with PTSD were lower than those in health subjects. They suggested that PTSD might be a neuroplastic disorder. This is an interesting study because several animal studies using an animal model of PTSD reported the decreased levels of BDNF in the brain including hippocampus. However, there are several issues to be clarified.

Results:

In this manuscript, the authors demonstrated that the blood levels of BDNF in patients with PTSD were lower than those in healthy subjects. However, numerous factors other than the diagnosis of PTSD affect the blood levels of BDNF. Although the authors mentioned that subgroup meta-analysis was not appropriate because of the small size of subjects in Discussion, it is required to demonstrate several factors that substantially affect the BDNF levels in Result.

As the authors mentioned in Discussion, it is well known that the blood BDNF levels in depression are lower. In this context, they have to show that how many papers assessed depressive mood in patients with PTSD in these 20 studies. If possible, please compare the blood BDNF levels between PTSD patients with and without depression.

As the authors mentioned, the polymorphism (Val66Met) of the BDNF gene affects the BDNF levels. So, they are required to show how many papers examined the genotype difference in the BDNF levels in these 20 studies.

It is well know that administration of SSRIs increased the blood levels of BDNF in patients with depression. Similarly, it is conceivable that SSRIs increase the blood BDNF levels in patients with PTSD. Thus, they should show the difference between unmedicated and medicated patients.

Discussion:

Although the authors did not demonstrate the influence of the genotype (Val66Met) on the BDNF levels in Results, they precisely discussed in Discussion. Because the section of the genotype in Discussion is redundant, please concisely discuss this issue.

As I mentioned in above, many factors affecting the blood BDNF levels were not examined in this study. In this context, the authors should provide the section of Limitation in Discussion in which they describe what factors are required to be examined to determine whether the blood levels of BDNF are involved in the pathophysiology of PTSD.

In addition, the authors did not show the ELISA kits used in these 20 studies. The study conducted by Placchini and colleagues in Scientific Reports (DOI: 10.1038/srep17989) reported that the 5 different kits exhibited very different inter-assay variations, and they identified two assays to obtain reliable measurements of human serum BDNF. Based on this finding, please mention the limitation of the BDNF assay in Limitation.

Reviewer #3: The authors performed a systematic review and meta-analysis on blood BDNF levels in patients with PTSD. I have the following comments:

1, the manuscript reported significant between-study heterogeneity. However, how the heterogeneity was calculated was not described. More importantly, the authors shoud use subgroup and meta-regression analyses to address the high levels of the between-study heterogeneity.

2, the authors may need to discuss the sampling source as an potential variable for the observed heterogeneity( reference see : Mol Psychiatry. 2017 Feb;22(2):312-320.).

3. Sensitivity analysis shoud be performed to demonstrate the robustness of the meta-analysis outcome

4. The authors described non-PTSD controls, were those healthy controls or disease controls or both included?

5. The language of the paper need to be thoroughly edited.

6. PLOS authors have the option to publish the peer review history of their article (what does this mean?). If published, this will include your full peer review and any attached files.

Reviewer #1: **Yes: **Bengt B. Arnetz

Reviewer #2: No

Reviewer #3: No

---

## [Author Response · Author response to Decision Letter 0]

8 Oct 2020

Additional Editor Comments

Comment 

I would like to suggest that the following articles should be discussed in the limitation of the discussion.

ELISA kits used in the study can recognize both BDNF and its precursor proBDNF because of lack of selectvity of antibodies used in the kits. Therefore, BDNF levels in the human blood are total BDNF and proBDNF. Please discuss the following articles.

Yoshida T, et al. PLOS ONE 2012; 7(8): e42676.

Hashimoto K. Eur Arch Psychiatry Clin Neurosci 2016; 266(3): 285-287.

Response

The following paragraph is added to the discussion session; “Standard ELISA kits lack proBDNF antibodies and are unable to distinguish between proBDNF and mature BDNF in the blood stream. New generations of ELISA kits provide antibodies for both pro and mature BDNF. In most included studies in the present meta-analysis, only BDNF was measured, except for the study by Aksu et al. who measured proBDNF in addition to BDNF and tPA. Aksu concluded that PTSD population had lower levels of both BDNF and its precursor, proBDNF compared to their healthy volunteers [8]. The possible limitation of the present study is the absence of proBDNF measures in the remaining studies rather than Aksu. Based on a prior investigations on the role of BDNF among people with major depressive disorder (MDD), although BDNF may be significantly lower in the disease group compared to the normal controls, the difference of proBDNF levels between MDD patients and healthy population was not significant [51].” 

Journal requirements

Comment 

Response

The manuscript, figures, tables, and references are carefully modified based on the PLOSOne guidelines. 

Comment 

https://www.sciencedirect.com/science/article/abs/pii/S0278584609001286?via%3Dihub

In your revision ensure you cite all your sources (including your own works), and quote or rephrase any duplicated text outside the methods section.

Further consideration is dependent on these concerns being addressed.

Response

We strictly checked the manuscript for overlap issues. 

Comment 

Please remove your figures from within your manuscript file, leaving your figures uploaded only as individual TIFF/EPS image files, uploaded separately. These will be automatically included in the reviewers’ PDF.

Response

Figures are removed from the manuscript. 

Comment 

Please include captions for your Supporting Information files at the end of your manuscript, and update any in-text citations to match accordingly. Please see our Supporting Information guidelines for more information: http://journals.plos.org/plosone/s/supporting-information

Response

Figure captions are added at the end of the manuscript. 

Review Comments to the Author

Reviewer #1

Comment 

This is an interesting and ambitious meta-analysis of the association between peripheral BDNF and PTSD. There are several, mostly methodological issues that need clarifications in order to properly assess the conclusions.

The authors need to define how the diagnosis of PTSD were determined in the meta-analysis. In the original 20 studies, there is a mixture of clinically diagnosed PTSD, and diagnosis based on scores on PTSD rating scales. For examples, (1) what cut-off scores were used in the various papers to classify a person a suffering from PTSD. How was exposure defined? To what degree were prior trauma, e g adverse childhood experiences accounted for? And, what scales were used, apart from clinical diagnostic interviews, to determine a person had met exposure criteria required at the time for the dx of PTSD. There is no info as to the time between the critical exposure event and the diagnosis of PTSD, nor time between diagnosis and the respective study date. Looking at the original studies, there peripheral concentration of BDNF various in both directions (higher/lower) comparing PTSD cases with controls. (2)In most cases, the 95% confidence interval overlaps with 1, that is, the risk ratio does not significantly differ from the null hypothesis. More info is needed as to how the NOS scores were assigned since these scores are critical because they appear to significantly change the weighting of studies and thus overall outcomes. The authors relate trauma exposure to increased neuroplasticity and memory consolidation. There is a need to expand the (3)Discussion section in terms of why one would expect lower levels of BDNF in PTSD patients - apart from referring to SNP and other pre-exposure changes. In terms of the hippocampus, well-controlled studies, e g involving twins where one has been exposed to trauma and develops PTSD, that there are no differences in hippocampal volume.

Overall, a potentially interesting study. However, the reader would benefit from more detailed information as to methodological approaches used in terms of a comparison across studies with very different design and study participants.

Response 

Many thanks for the precise comments of yours, which helped improving the manuscript significantly. We have already included several subgroup meta-analyses, meta-regression, and sensitivity analyses. Meta-analyses of BDNF levels were performed to compare PTSD patients and non-PTSD controls. There were lower levels of BDNF in PTSD patients than in healthy subjects (SMD = -0.57; 95% confidence interval: -1.02 to -0.12) (Figure 2). However, PTSD patients and controls without PTSD did not differ in BDNF levels. Subgroup meta-analyses demonstrated neither effect of the sample (plasma and serum) nor the technique (ELISA and sandwich ELISA) used for BDNF measurement (Figures 3 and 4). Also, meta-regression showed no significant effect of age, gender, NOS, and sample size. By specifying I2 = 10% and tau2 = 0.25, sensitivity meta-analyses were also performed to check the impact of heterogeneity on effect size. In both cases, the effect size was significant (tau2 = 0.25: Hedges’s g, 0.53, p = 0.000; I2 = 10%: Hedges’s g, 0.57, p = 0.000). No evidence of publication bias was found (Figure 5; Egger’s p = 0.567; Begg’s p = 0.284). 

Also, we have thoroughly revised the manuscript to address the issues and concerns you kindly shared with us. The text highlighted yellow reflects changes. 

(1) Neuroinflammation is known to affect several BDNF-related signaling pathways causing different brain pathologies. Aging is among the most affected brain changes by the down regulation of neurotropic factors. BDNF is also found to be associated in the pathogenesis of several neuropsychiatric and neurodegenerative disorders including major depressive disorder (MDD), bipolar disorder (BD), and schizophrenia, Parkinson disease, Alzheimer disease, and epilepsy. It is expected that presenting a particular mental illness relies on the combination of environmental, genetic, and temperamental factors.

(2) In regards to the underlying depression as a confounder variable, attitudes were diverse in our selected studies. Some papers excluded subjects with depression to dismiss the potential effect of this disorder on the BDNF level. However, others either did not exclude the depressed samples or specifically investigated the correlation of BDNF levels in PTSD patients in a MDD population. In the study by Aksu et al., measured levels of BDNF, proBDNF, and tPA showed no difference between PTSD group with depressive symptoms (21%) and the remaining PTSD population. Although 46 out of 83 cases (55%) in Blessing et al.’s study had a confirmed diagnosis of MDD in addition to the PTSD and twenty of them consume antidepressant drugs, no correlation analysis was carried out to address the effect of this factors on BDNF measures. Despite detecting symptoms of depression, mania, and anxiety with or without drug consumption, Bucker et al and Dotta-Panichi et al. did not include these variables in their analysis. 

On the other hand, recurrent MDD patients were investigated to address the association of BDNF level and trauma exposure in the presence of MDD as a major comorbidity by Grassi-Oliveira et al. They concluded that patients with MDD had lower levels of BDNF compared to the healthy control population and within the MDD group the ones with the presence of childhood physical neglect (CPN), as a predisposing factor for PTSD, observed to have even lower BDNF measures. 

(3) Three papers evaluated the effect of genetic polymorphism on the BDNF plasma levels in the PTSD setting. In the study by van den Heuvel et al. Met allele was associated with higher incidence of acute stress disorder (ASD), while after six months follow-up the prevalence of PTSD was lower in the Met allele cases versus the Val allele group (one and 14 patients respectively); although none of the observed alterations were statistically significant. Zhang et al. showed that patients with a Met/Met genotype had a higher PTSD symptom severity, while Val carriers had lower PTSD symptom severity. In addition Met carriers had a higher level of BDNF levels in the PTSD group. Lastly, zero association was noted in regards to BDNF G11757 and rs6265 polymorphisms and PTSD severity. 

(4) Although 52.9% of MDD with CPN group and 41.2% of MDD without CPN group consumed selective serotonin reuptake inhibitors (SSRI) as antidepressant agent, the study did not control the outcome to address this issue. Antidepressant consumption, current or past psychiatric disorder or depressive symptoms had zero effect on BDNF level in the sample studied by Hauk et al. Ultimately, in the study by Yang et al. a conclusion was made that women with comorbid PTSD-depression have 1.52-fold increased odds of having lower levels of serum BDNF compared to the women with neither of these conditions. Nevertheless, the difference between BDNF measures for the two mentioned group did not reach a level of significance despite the lower BDNF amount in the PTSD-depression group. Despite SSRI or other psychoactive drug consumption in some of our included studies, the effect of medication was not analyzed in any of the studies.

(5) One major limitation to our study is that all the included papers evaluated the BDNF levels in peripheral blood stream which is still unknown to what extend the peripheral levels correspond to the CNS levels of BDNF. There are many confounding factors that can potentially alter levels of BDNF regardless of traumatic experience. MDD, bipolar, SSRI consumptions are the common instances which should be considered in the future study designs. Almost all the studies, except one, did not measure proBDNF as a potentially involved biomarker in the disease course. BDNF gene polymorphism should also be considered in the future study designs.

(6) Several ELISA kits are used in this 20 studies, the majority of the included studies used Quantikine ELISA kit (7 studies). Chemicon, Promega, Mallipore ChemiKine and Milliplex, Picokine, Raybiotech, Human BDNF ELISA, and Nanjing Jiancheng ELISA kits for human BDNF were the other used kits in the studies. Among which Mallipore Milliplex had the widest range of detection (12–50000 pg/ml) whereas Promega had a close range of BDNF detection (7.8–500 pg/ml). In addition Mallipore Milliplex was the most sensitive kit among all used with detecting 2.5 pg/ml of BDNF. Lastly Mallipore Milliplex and Quantikine from R&D were the top two kits in regards of higher intra-assay CV and inter-assay CV among all.

Reviewer #2

The authors examined the difference in the blood levels of BDNF between patients with PTSD and healthy subjects using the meta-analysis. They showed that the BDNF levels in patients with PTSD were lower than those in health subjects. They suggested that PTSD might be a neuroplastic disorder. This is an interesting study because several animal studies using an animal model of PTSD reported the decreased levels of BDNF in the brain including hippocampus. However, there are several issues to be clarified.

Comment 

In this manuscript, the authors demonstrated that the blood levels of BDNF in patients with PTSD were lower than those in healthy subjects. However, numerous factors other than the diagnosis of PTSD affect the blood levels of BDNF. Although the authors mentioned that subgroup meta-analysis was not appropriate because of the small size of subjects in Discussion, it is required to 1. demonstrate several factors that substantially affect the BDNF levels in Result;

2. As the authors mentioned in Discussion, it is well known that the blood BDNF levels in depression are lower. In this context, they have to show that how many papers assessed depressive mood in patients with PTSD in these 20 studies. If possible, please compare the blood BDNF levels between PTSD patients with and without depression;

3. As the authors mentioned, the polymorphism (Val66Met) of the BDNF gene affects the BDNF levels. So, they are required to show how many papers examined the genotype difference in the BDNF levels in these 20 studies;

4. It is well know that administration of SSRIs increased the blood levels of BDNF in patients with depression. Similarly, it is conceivable that SSRIs increase the blood BDNF levels in patients with PTSD. Thus, they should show the difference between unmedicated and medicated patients.

5. Discussion: Although the authors did not demonstrate the influence of the genotype (Val66Met) on the BDNF levels in Results, they precisely discussed in Discussion. Because the section of the genotype in Discussion is redundant, please concisely discuss this issue.

As I mentioned in above, many factors affecting the blood BDNF levels were not examined in this study. In this context, the authors should provide the section of Limitation in Discussion in which they describe what factors are required to be examined to determine whether the blood levels of BDNF are involved in the pathophysiology of PTSD; and

6. In addition, the authors did not show the ELISA kits used in these 20 studies. The study conducted by Placchini and colleagues in Scientific Reports (DOI: 10.1038/srep17989) reported that the 5 different kits exhibited very different inter-assay variations, and they identified two assays to obtain reliable measurements of human serum BDNF. Based on this finding, please mention the limitation of the BDNF assay in Limitation.

Response 

We appreciate your kind comments in order to improve our text. We sharing the response to your comments here under and you can also find them in the actual manuscript: 

(1) Neuroinflammation is known to affect several BDNF-related signaling pathways causing different brain pathologies. Aging is among the most affected brain changes by the down regulation of neurotropic factors. BDNF is also found to be associated in the pathogenesis of several neuropsychiatric and neurodegenerative disorders including major depressive disorder (MDD), bipolar disorder (BD), and schizophrenia, Parkinson disease, Alzheimer disease, and epilepsy. It is expected that presenting a particular mental illness relies on the combination of environmental, genetic, and temperamental factors.

(2) In regards to the underlying depression as a confounder variable, attitudes were diverse in our selected studies. Some papers excluded subjects with depression to dismiss the potential effect of this disorder on the BDNF level. However, others either did not exclude the depressed samples or specifically investigated the correlation of BDNF levels in PTSD patients in a MDD population. In the study by Aksu et al., measured levels of BDNF, proBDNF, and tPA showed no difference between PTSD group with depressive symptoms (21%) and the remaining PTSD population. Although 46 out of 83 cases (55%) in Blessing et al.’s study had a confirmed diagnosis of MDD in addition to the PTSD and twenty of them consume antidepressant drugs, no correlation analysis was carried out to address the effect of this factors on BDNF measures. Despite detecting symptoms of depression, mania, and anxiety with or without drug consumption, Bucker et al and Dotta-Panichi et al. did not include these variables in their analysis. 

On the other hand, recurrent MDD patients were investigated to address the association of BDNF level and trauma exposure in the presence of MDD as a major comorbidity by Grassi-Oliveira et al. They concluded that patients with MDD had lower levels of BDNF compared to the healthy control population and within the MDD group the ones with the presence of childhood physical neglect (CPN), as a predisposing factor for PTSD, observed to have even lower BDNF measures. 

(3) Three papers evaluated the effect of genetic polymorphism on the BDNF plasma levels in the PTSD setting. In the study by van den Heuvel et al. Met allele was associated with higher incidence of acute stress disorder (ASD), while after six months follow-up the prevalence of PTSD was lower in the Met allele cases versus the Val allele group (one and 14 patients respectively); although none of the observed alterations were statistically significant. Zhang et al. showed that patients with a Met/Met genotype had a higher PTSD symptom severity, while Val carriers had lower PTSD symptom severity. In addition Met carriers had a higher level of BDNF levels in the PTSD group. Lastly, zero association was noted in regards to BDNF G11757 and rs6265 polymorphisms and PTSD severity. 

(4) Although 52.9% of MDD with CPN group and 41.2% of MDD without CPN group consumed selective serotonin reuptake inhibitors (SSRI) as antidepressant agent, the study did not control the outcome to address this issue. Antidepressant consumption, current or past psychiatric disorder or depressive symptoms had zero effect on BDNF level in the sample studied by Hauk et al. Ultimately, in the study by Yang et al. a conclusion was made that women with comorbid PTSD-depression have 1.52-fold increased odds of having lower levels of serum BDNF compared to the women with neither of these conditions. Nevertheless, the difference between BDNF measures for the two mentioned group did not reach a level of significance despite the lower BDNF amount in the PTSD-depression group. Despite SSRI or other psychoactive drug consumption in some of our included studies, the effect of medication was not analyzed in any of the studies.

(5) One major limitation to our study is that all the included papers evaluated the BDNF levels in peripheral blood stream which is still unknown to what extend the peripheral levels correspond to the CNS levels of BDNF. There are many confounding factors that can potentially alter levels of BDNF regardless of traumatic experience. MDD, bipolar, SSRI consumptions are the common instances which should be considered in the future study designs. Almost all the studies, except one, did not measure proBDNF as a potentially involved biomarker in the disease course. BDNF gene polymorphism should also be considered in the future study designs.

(6) Several ELISA kits are used in this 20 studies, the majority of the included studies used Quantikine ELISA kit (7 studies). Chemicon, Promega, Mallipore ChemiKine and Milliplex, Picokine, Raybiotech, Human BDNF ELISA, and Nanjing Jiancheng ELISA kits for human BDNF were the other used kits in the studies. Among which Mallipore Milliplex had the widest range of detection (12–50000 pg/ml) whereas Promega had a close range of BDNF detection (7.8–500 pg/ml). In addition Mallipore Milliplex was the most sensitive kit among all used with detecting 2.5 pg/ml of BDNF. Lastly Mallipore Milliplex and Quantikine from R&D were the top two kits in regards of higher intra-assay CV and inter-assay CV among all.

Reviewer #3

Comment

The authors performed a systematic review and meta-analysis on blood BDNF levels in patients with PTSD. I have the following comments:

1. the manuscript reported significant between-study heterogeneity. However, how the heterogeneity was calculated was not described. More importantly, the authors shoud use subgroup and meta-regression analyses to address the high levels of the between-study heterogeneity.

2. the authors may need to discuss the sampling source as an potential variable for the observed heterogeneity (reference see : Mol Psychiatry. 2017 Feb;22(2):312-320.).

3. Sensitivity analysis shoud be performed to demonstrate the robustness of the meta-analysis outcome

4. The authors described non-PTSD controls, were those healthy controls or disease controls or both included?

5. The language of the paper need to be thoroughly edited.

Response

Many thanks for your comment, which was constructive. Subgroup meta-analysis and meta-regression are already included in the revision. Meta-analyses of BDNF levels were performed to compare PTSD patients and non-PTSD controls. There were lower levels of BDNF in PTSD patients than in healthy subjects (SMD = -0.57; 95% confidence interval: -1.02 to -0.12) (Figure 2). However, PTSD patients and controls without PTSD did not differ in BDNF levels. Subgroup meta-analyses demonstrated neither effect of the sample (plasma and serum) nor the technique (ELISA and sandwich ELISA) used for BDNF measurement (Figures 3 and 4). Also, meta-regression showed no significant effect of age, gender, NOS, and sample size. By specifying I2 = 10% and tau2 = 0.25, sensitivity meta-analyses were also performed to check the impact of heterogeneity on effect size. In both cases, the effect size was significant (tau2 = 0.25: Hedges’s g, 0.53, p = 0.000; I2 = 10%: Hedges’s g, 0.57, p = 0.000). No evidence of publication bias was found (Figure 5; Egger’s p = 0.567; Begg’s p = 0.284). 

We thoroughly went through the manuscript. We hope we could address your concerns adequately.

---

## [Decision Letter · Decision Letter 1]

22 Oct 2020

PONE-D-20-17402R1

Peripheral Blood Levels of Brain-Derived Neurotrophic Factor in Patients with Post-traumatic Stress Disorder (PTSD): A Systematic Review and Meta-analysis

PLOS ONE

Dear Dr. Rezaei,

Thank you for submitting your manuscript to PLOS ONE. After careful consideration, we feel that it has merit but does not fully meet PLOS ONE’s publication criteria as it currently stands. Therefore, we invite you to submit a revised version of the manuscript that addresses the points raised during the review process.

Although you addressed to reviewer's comments, you did not respond my comments carefully. Please revise your manuscript again.

We look forward to receiving your revised manuscript.

Kind regards,

Kenji Hashimoto, PhD

Academic Editor

PLOS ONE

Additional Editor Comments (if provided):

Although you addressed all comments from three reviewers, you did not respond my comment. Commercially available BDNF ELISA kits can recognize both BDNF and its precursor proBDNF. Therefore, many reports show total values of BDNF (mature form) and proBDNF in the human blood. In the limitation section, the authors should add some sentences including references suggested by the editor.

Reviewers' comments:

Reviewer's Responses to Questions

**Comments to the Author**

1. If the authors have adequately addressed your comments raised in a previous round of review and you feel that this manuscript is now acceptable for publication, you may indicate that here to bypass the “Comments to the Author” section, enter your conflict of interest statement in the “Confidential to Editor” section, and submit your "Accept" recommendation.

Reviewer #1: All comments have been addressed

Reviewer #2: All comments have been addressed

Reviewer #3: (No Response)

2. Is the manuscript technically sound, and do the data support the conclusions?

Reviewer #1: Yes

Reviewer #2: Yes

Reviewer #3: (No Response)

3. Has the statistical analysis been performed appropriately and rigorously? 

Reviewer #1: Yes

Reviewer #2: Yes

Reviewer #3: (No Response)

4. Have the authors made all data underlying the findings in their manuscript fully available?

Reviewer #1: Yes

Reviewer #2: Yes

Reviewer #3: (No Response)

5. Is the manuscript presented in an intelligible fashion and written in standard English?

Reviewer #1: Yes

Reviewer #2: Yes

Reviewer #3: (No Response)

6. Review Comments to the Author

Reviewer #1: The authors have carefully addressed my critique in the revised version of the manuscript . I have no further comments.

Reviewer #2: The authors appropriately responded to all comments.

In addition, DNA methylation status of the BDNF gene will be attractive.

Reviewer #3: (No Response)

7. PLOS authors have the option to publish the peer review history of their article (what does this mean?). If published, this will include your full peer review and any attached files.

Reviewer #1: **Yes: **Bengt B. Arnetz, MD, PhD, MPH, MScEpi

Reviewer #2: No

Reviewer #3: No

---

## [Author Response · Author response to Decision Letter 1]

22 Oct 2020

PONE-D-20-17402R1

Peripheral Blood Levels of Brain-Derived Neurotrophic Factor in Patients with Post-traumatic Stress Disorder (PTSD): A Systematic Review and Meta-analysis

PLOS ONE

Dear Prof. Hashimoto,

We are pleased to submit the second version of a systematic review and meta-analysis type of article entitled “Peripheral Blood Levels of Brain-Derived Neurotrophic Factor in Patients with Post-traumatic Stress Disorder (PTSD): A Systematic Review and Meta-analysis” for publication in the PLOS ONE. 

Many thanks for your constructive suggestion which is now incorporated into the revision.

We hope to see it published soon 

Kind regards

Nima Rezaei, MD, PhD

Professor of Clinical Immunology

Founding President of USERN

Deputy President of Research Center for Immunodeficiencies

Vice Dean of International Affairs, School of Medicine,

Tehran University of Medical Sciences

Children’s Medical Center Hospital,

Dr. Gharib St, Keshavarz Blvd,

Tehran, Iran

Tel: +9821-6657-6573

Fax: +9821-6692-9235

E-mail: rezaei_nima@tums.ac.ir

http://usern.tums.ac.ir/User/CV/rezaei_nima

Additional Editor Comments

Comment 

Although you addressed all comments from three reviewers, you did not respond my comment. Commercially available BDNF ELISA kits can recognize both BDNF and its precursor proBDNF. Therefore, many reports show total values of BDNF (mature form) and proBDNF in the human blood. In the limitation section, the authors should add some sentences including references suggested by the editor.

Response

Dear Professor Hashimoto

Many thanks for your constructive suggestion.

We therefore considered the issue as follows: “It is noteworthy to point out that ELISA kits mostly can consider mature BDNF and proBDNF as distinct in a limited sense [71]. Using new ELISA kits which can measure mature BDNF with higher specificity is recommended as another line of future research [72].” 

Yoshida T, et al. PLOS ONE 2012; 7(8): e42676.

Hashimoto K. Eur Arch Psychiatry Clin Neurosci 2016; 266(3): 285-287.

---

## [Editor Report · Decision Letter 2]

23 Oct 2020

Peripheral Blood Levels of Brain-Derived Neurotrophic Factor in Patients with Post-traumatic Stress Disorder (PTSD): A Systematic Review and Meta-analysis

PONE-D-20-17402R2

Dear Dr. Rezaei,

We’re pleased to inform you that your manuscript has been judged scientifically suitable for publication and will be formally accepted for publication once it meets all outstanding technical requirements.

Kind regards,

Kenji Hashimoto, PhD

Section Editor

PLOS ONE

Additional Editor Comments (optional):

My comments have been addressed.
---

## [Editor Report · Acceptance letter]

27 Oct 2020

PONE-D-20-17402R2 

Peripheral blood levels of brain-derived neurotrophic factor in patients with post-traumatic stress disorder (PTSD): a systematic review and meta-analysis 

Dear Dr. Rezaei:

I'm pleased to inform you that your manuscript has been deemed suitable for publication in PLOS ONE. Congratulations! Your manuscript is now with our production department. 

Kind regards, 

on behalf of

Prof. Kenji Hashimoto 

Section Editor

PLOS ONE